# Solving Ecological Problems through Physical Computing to Ensure Gender Balance in STEM Education

Gabrielė Stupurienė [1,*], Tatjana Jevsikova [1,2] and Anita Juškevičienė [1,2]

1 Institute of Educational Sciences, Faculty of Philosophy, Vilnius University, Universiteto Str. 9, LT-01513 Vilnius, Lithuania; tatjana.jevsikova@mif.vu.lt (T.J.); anita.juskeviciene@mif.vu.lt (A.J.)
2 Institute of Data Science and Digital Technologies, Vilnius University, Akademijos Str. 4, LT-08412 Vilnius, Lithuania
* Correspondence: gabriele.stupuriene@mif.vu.lt

**Abstract:** Research and practice have shown that female students are less interested in engineering and programming. This is related to gender stereotypes and technological self-efficacy. Research has also pointed out that students in rural schools tend to do less well in STEM subjects and are less likely to pursue STEM studies than their peers from large cities. Previous studies have highlighted the benefits of hands-on real-world-related engineering projects by building connections with students' interests and technology while giving them something exciting to focus on. This study is aimed at investigating whether and how students' individual characteristics (such as attitudes toward engineering and technology, motivation, and technology anxiety) are associated with rural school students' engagement, gender differences, and inclusion in sustainable ecological engineering activities with Arduino microcontrollers. Surveys were conducted before and after the activity with pupils of a rural lower secondary school (ages 13–15). The results show that, female students' initial attitude toward engineering and technology was significantly less positive than that of male students. Despite being novices in physical computing, a whole group of pupils were intrinsically motivated while performing these activities. The findings of this study provide transferable insights into practical STEM education that may strengthen students' engagement, motivation, and achievement in STEM. The implications of the results of this study can be useful for a better understanding of the individual factors of students that influence future engineering activity design and STEM career selection opportunities.

**Keywords:** STEM education; inclusion and gender balance; Arduino microcontrollers; hands-on activities; physical computing; situational motivation; engineering and technology attitude; technology anxiety





## 1. Introduction

In many professional STEM (science, technology, engineering, and math) fields, the majority of employees are men, especially in the technical fields of STEM, such as computer science and engineering. It is still a challenge to change gender stereotypes regarding these traditionally male fields. In order to change children's attitudes about computer science and engineering, a variety of educational initiatives for girls have emerged, such as short- and long-term programs, projects, and summer schools [1–3].

The participation of women in technical fields is very low compared to that of men. This is due to many factors—for example, stereotypes or anxiety about confirming a negative stereotype about one's performance of a task or activity [4]. A misconception about engineering degrees and engineering is that "engineering is not a female profession" [5]. Some researchers have proposed the introduction of engineering from early ages in the formative period for attitudes and interest in engineering in children (5–7 years old) [4]. However, it is never too late to start, and such professions can become attractive even in

tertiary education. It is important that educational institutions' authorities, teachers, and parents present and describe such disciplines by promoting a positive impact on cultural knowledge about gender and professional careers, making the idea of entering engineering attractive for women. This can be done by involving girls in engineering-related projects, supporting initiatives for girls enrolling in STEM, and presenting examples of successful female engineers. This will help to move from having interest in science to acquiring the skills, knowledge, self-efficiency, and aspiration for a career in science [1].

It is important to develop an understanding of STEM from an early age by incorporating it into curricula and changing pedagogies [1]. Another very important factor in causing girls to be interested in science is the opportunity to "open a black box of technology" and to help girls and boys better understand what engineering and technological development are [3].

As presented earlier, researchers have proposed many solutions in order to minimize gender segregation across engineering. Additionally, it is important to highlight that women usually choose professions that are useful for the community. Thus, it is important to emphasize a sense of community utility in STEM fields [5] and opportunities to serve society. In addition, because women think that engineering is too complicated and too risky for their professional expectations, it is important to provide more knowledge about engineering degrees and engineering professions among women. However, despite the fact that females view engineering professions as socially valued, there are other issues: the pay gap between men and women and the differences in positions held. The tendencies show that women's wages and positions (both rank and promotions) are lower [6].

People vary; some prefer science and technology and some prefer theory and practice, as the aims are to understand principles and be able to put them practice, to use technologies or develop them, and to solve real-life problems [3]. Of course, not everyone can be (and must be) interested in STEM, but the modern citizens of the digital age must know the engineering side of technology and the principles of operation in order to easily reap the benefits that they provide. Modern pupils would like to see the results here and now, to try to develop products, and to use creativity, and they should not be afraid to fail. If pupils fail in the first stages—for example, on an abstract level—assistance could be provided to them by choosing other representations of the same knowledge. In such a way, learners are motivated by their own creativity while implementing tangible microcontroller-based projects [7].

Educators and researchers widely agree that learners are motivated and attracted to the learning process by project implementation (hands-on activities) [8,9]. Such projects and practical applications often refer to physical computing and the making-and-tinkering paradigm, which brings computer science and engineering concepts off the screen and into the real world so that the students can interact with them [10].

In this research, we introduced problems of sustainable ecology (smart greenhouses) to students from a rural lower secondary school (grade 7–8). Activities with Arduino microcontrollers were chosen, as they were designed to engage children in several different aspects of engineering: building, programming, designing, and iterative testing and re-design [11,12]. They had a positive effect on students' academics and their understanding of engineering concepts [13], and they also fostered students' creativity and a sense of discovery [14], which are important problem-solving skills [15]. Arduino was chosen based on its functionality, suitability for the design of the activity, availability, and previous positive experience in Arduino-based teaching activities. The Arduino prototyping board is considered to be an attractive physical device; it does not require much time to learn how to use it or how to configure it, and it is, therefore, becoming increasingly popular among researchers and educators. In fact, the selection of a particular model of microcontroller is not essential, as the main goal is to understand the basic principles of microcontroller programming that help to solve the problem posed. When the main principles are learned, the learning can be transferred and applied with another technological tool.

It is important to point out that our activities at the rural school were led by a female teaching team that had previous experience in attracting girls into the STEM field by using activities with Arduino. Sullivan and Bers's [16] study showed preliminary evidence that female robotics instructors may have a more positive impact on girls' performance on robotics tasks. Female mentor role models are seen as one of the promising ways to reduce the gender gap in STEM [17].

### 1.1. Background and Motivation

According to Murphy [18], an important but often overlooked issue in STEM education is the relatively low engagement and poor performance of rural school students in STEM education. Students in rural schools tend to do less well in STEM subjects and are less likely to pursue STEM studies than their peers from large cities. Considering this point, we chose a rural school for our study.

The first step in finding out why STEM is not attractive to women is to find out the prevailing gender stereotypes. To make science interesting for girls, it is not enough to break down those stereotypes. Students' expectations of success are closely linked to their self-concept (beliefs about their abilities in a given domain) and their self-efficacy (beliefs about their ability to perform a given task) [19]. Thus, it is important to clarify what individual factors determine interest and success in STEM.

Based on previous work [20] and a literature review, we identified the main individual characteristics that can impact the increase in interest in STEM, and the following were used in our survey: motivation, attitude toward engineering and technology, technology anxiety, and behavioral intention. Learning, as with any other human behavior, is underlined by different types of motivation [21]. According to self-determination theory [22], behaviors driven by intrinsic motivation are those that are engaged in for their own sake, i.e., what one wants to do, with a sense of freedom to choose, while extrinsic motivation is related to various external regulations. Different types of motivation are related to various types of outcomes. Intrinsic motivation is mostly associated with positive outcomes (e.g., persistence) [23]. The motivation that students experience when engaging in a particular activity (e.g., programming Arduino microcontrollers) is called situational motivation [23]. In our study, we measured students' intrinsic and extrinsic situational motivation in relation to a microcontroller activity for a smart greenhouse solution. For the acceptance of and involvement in microcontroller activities, students' initial attitude toward engineering and technology is very important. This attitude is understood as students' self-efficacy related to engineering and technology, as well as expectations for future value gained from success in this field [24]. Technology anxiety is a term derived from computer anxiety [25] and is understood as a negative emotional response; it describes an individual's perceived apprehension or discomfort related to using a technology [26]. Many female students think that engineering is a difficult area, as it requires technology uptake skills [27]. In our study, technology anxiety is referred to in two dimensions: as general computer anxiety before activities and microcontroller anxiety after the activities. An individual's tendency to perform a behavior is described by behavioral intention [28]. This is a core construct of the theory of planned behavior, which represents motivational factors that impact actual behavior [29]. Behavioral intention is studied as an outcome variable in technology acceptance models and is used as a predictor of technology acceptance behavior [30]. It was shown in previous studies that behavioral intention was significantly positively associated with actual students' behavior of STEM integration [31]. In our study, behavioral intention refers to students' willingness to use microcontrollers in their learning activities.

### 1.2. Aim and Research Questions

The aim of this research is to investigate whether and how students' individual characteristics are associated with their engagement with and the gender differences and inclusion in sustainable ecological engineering activities with Arduino microcontrollers.

For this purpose, the following research questions are considered:

RQ1. How are students' attitudes toward engineering and technology, intrinsic and extrinsic motivation, technology anxiety, behavioral intention to use microcontrollers, and attitude toward future physical computing activities inter-correlated?

RQ2. What are the differences between groups of female and male students in terms of the individual characteristics (such as attitudes toward engineering and technology, motivation, technology anxiety) studied?

The paper is structured as follows: Section 2 identifies the material of the proposed physical computing activities in the school and the research methodology. The results of the analysis of the collected data are presented in Section 3. Finally, we discuss our findings and provide directions for future research and practical implications.

## 2. Materials and Methods

### 2.1. Methodology for the Implementation of Microcontroller Activities

The smart greenhouse is a solution for sustainable agriculture and ecological issues—for example, as an ecosystem managed by humans to allow the growth of plants in climatic zones or periods of the year when this would not be possible in a natural environment. This shows how humans are able to intervene in the balance of our planet in a more or less sustainable way. A realistic context was proposed for students as a way to stimulate an interdisciplinary path that helps students develop a sensitivity for and concrete awareness of issues relating to the environment, global development, and the conservation of life on our planet. The context is also relevant for students in schools in rural areas, such as where our study was held. In addition, the greenhouse topic is appealing to both girls and boys, but especially for girls because it is related to opportunities to help society.

During a two-hour activity at a rural school in northwest Lithuania in January 2022, 7th- and 8th-grade students (13 to 15 years old) had four projects to implement with the aim of detecting the temperature and humidity of the air, the concentration of a gas (carbon dioxide), and the brightness level inside a greenhouse and to control a light with a mobile device (Appendix A). The main Arduino-kit-based activities were aimed at exploring smart greenhouse solutions for the comfortable cultivation of plants. The mini-projects were implemented by groups of 2–3 boys and girls (Figure 1). Students were informed in advance about the survey in which they were taking part and that their participation was voluntary.

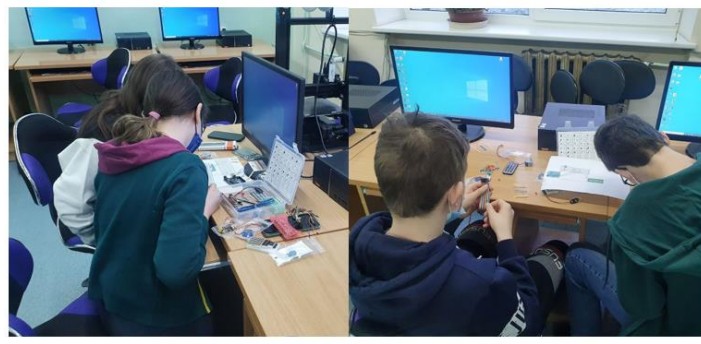

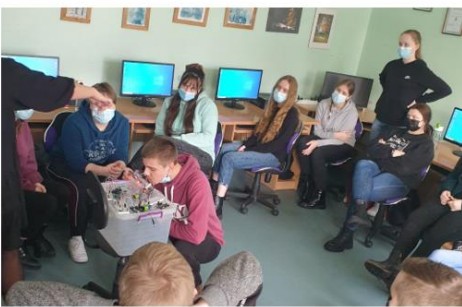

**Figure 1.** Working in groups with Arduino microcontrollers and demonstration of the results.

The initial idea was to use Tinkercad for the greenhouse modeling before using Arduino, but there was no possibility to do so because Tinkercad lacked several of the necessary libraries and sensors. On the other hand, the students just came back from distance learning, so, from a practical point of view, they were lacking the opportunity for real simulations and the possibility to see how the sensors worked in a real environment.

The Arduino prototypes for these projects required students to use abstraction and logical thinking skills. Abstraction was needed during the reading and understanding of schemas; logical thinking was needed during the assembly of electrical circuits. Other computational thinking skills were also required during the activities, such as those of algorithms, debugging, and simulation. The class instructors changed the codes slightly to adapt them to the classroom environment, as different rooms had different temperatures and light levels. In addition, to test if the prototypes worked correctly, students were required to simulate the darkness, temperature, and humidity levels, as well as gas concentration changes. The skill of creativity was encouraged during the project for the control of an LED light via a mobile app. There was a great deal of interest among the students in programming a light-control interface on a smartphone with the MIT App Inventor.

After successfully completing their projects and simulations, the teams presented their projects to the whole class, reflecting on what the greenhouse effect might mean and discussing the topic of energy resources. In addition to providing knowledge on how to design, build, and code simple intelligent solutions and automated systems that address the problems embedded in the project, such activities also encouraged students to think about how to develop a concrete awareness of issues regarding global development and environmental sustainability.

The school activities were based on these working principles:

1.  Reducing theoretical assumptions and inputs as much as possible at the very beginning;
2.  Empowering students to get the chance to learn the fundamentals of physical computing in integration with STEM subjects;
3.  Partial failure is an important part of scientific work that is followed by understanding mistakes and gaining a tolerance for frustration;
4.  Encouraging students to discuss and make reflections.

### 2.2. The Survey and Instruments

In order to collect data on students' individual characteristics, a questionnaire was developed to see how much the students agreed or disagreed with statements about engineering and microcontroller activities. The questionnaire consisted of sets of questions (statements) that were delivered before and after the activities. The following instruments were used:

*   S-STEAM scale: Construct of Engineering and Technology Attitudes consisting of 9 items about students' self-efficacy related to engineering and technology and expectations for future value gained from success in this field measured on a 5-item Likert scale [24]. The items were accompanied with a paragraph defining the engineering and technology field for students. These items were presented to the students before the activity.
*   The Situational Motivation Scale (SIMS): Intrinsic and extrinsic motivation constructs consisting of 4 items each, which were measured on a 6-item scale [23]. Students rated these items after they completed the activity.
*   Technology anxiety: A 4-item scale adapted from Saadé and Kira [32] was used at two points in time. Before the activity, technology anxiety was measured as computer anxiety (as students might not have been familiar with microcontrollers yet); after the activity, the 4 items were adapted to measure microcontroller anxiety.
*   Behavioral intention: Three items were adapted for the microcontroller context from Venkatesh et al. [30] and delivered to the students after the activity. They were measured on a 5-point Likert scale.

- Attitude toward future microcontroller activities consisted of 2 items, which were rated on a 5-point Likert scale: "I would like to have more similar activities in the future", and "I would like to learn more about how to program microcontrollers".

After the activities were completed, students were asked to self-evaluate their activity results (group projects) on a 5-point scale. In addition, the questionnaires contained questions for collecting demographic information and encoding instructions in order to match the questionnaires for the same students before and after the activity.

### 2.3. Data Analysis

Quantitative analysis methods were used to analyze the collected data. As we worked with a small sample and cannot assume normality, distribution-free non-parametric measures were utilized for testing of the hypotheses:

- Spearman's rank correlations were used in order to test the relationships between the pairs of constructs.
- The Mann–Whitney U test was used to compare differences between two independent samples.
- The Wilcoxon signed-rank test was used to compare differences between two related samples.

The significance level was set to 5%. In some cases, 10% significance was also accepted, and this level is clearly stated in the description of the results. For the statistical analysis, we used the IBM SPSS Statistics 26 software package.

## 3. Results

In this section, we report the results of the quantitative analysis of this study.

### 3.1. Demographic Information

In total, 31 students (61.3% male, 38.7% female) of school grades 7 and 8 participated in the study (Table 1). The age of the respondents ranged from 13 to 15 years, with a median equal to 13. The vast majority of the participants were new to microcontrollers: only two students out of the 31 reported having earlier experience in using microcontrollers (one female student and one male student).

**Table 1.** Demographic characteristics of the participants.

| Gender | N | % |
|---|---|---|
| Female | 12 | 38.7 |
| Male | 19 | 61.3 |
| Total | 31 | 100 |
| **Age, years** | N | % |
| 13 | 19 | 41.9 |
| 14 | 11 | 35.5 |
| 15 | 1 | 3.2 |
| Total | 31 | 100 |
| **School grade** | N | % |
| 7th | 24 | 77.4 |
| 8th | 7 | 22.6 |
| Total | 31 | 100 |
| **Experience** | N | % |
| Had used microcontrollers | 2 | 6.5 |
| Had never used microcontrollers before | 29 | 93.5 |
| **Total** | 31 | 100 |

### 3.2. Relationships between Constructs

In order to examine the relationships between the constructs measured in this study, we calculated the scores for each construct as a sum of the ratings for each statement of

that construct. The ranges of scores and the descriptive statistics for each construct are presented in Table 2.

**Table 2.** Descriptive statistics for the measured constructs.

| | Range of Scores | Minimum | Maximum | Mean | Std. Deviation |
|---|---|---|---|---|---|
| Engineering and technology (ET) | 9–45 | 16 | 45 | 32.13 | 5.954 |
| Motivation (intrinsic) (MI) | 4–28 | 12 | 28 | 22.03 | 4.680 |
| Motivation (extrinsic) (ME) | 4–28 | 6 | 28 | 16.94 | 6.282 |
| Computer anxiety (pre) (CA) | 4–20 | 4 | 18 | 10.84 | 3.513 |
| Microcontroller anxiety (post) (MA) | 4–20 | 4 | 17 | 9.74 | 4.033 |
| Behavioral intention (BI) | 3–15 | 5 | 15 | 10.42 | 2.514 |
| Attitude (A) | 2–10 | 2 | 10 | 7.42 | 2.046 |
| Self-evaluation of the result (SR) | 1–5 | 1 | 5 | 3.74 | 1.316 |

Pair relationships between constructs were explored using Spearman's rank correlations (Table 3).

**Table 3.** Spearman's rank correlation matrix for the constructs used in the study.

| | | ET | CA | MA | MI | ME | BI | A | SR |
|---|---|---|---|---|---|---|---|---|---|
| ET | ρ | – | | | | | | | |
| | Sig. | – | | | | | | | |
| CA | ρ | 0.072 | – | | | | | | |
| | Sig. | 0.698 | – | | | | | | |
| MA | ρ | 0.020 | 0.717 ** | – | | | | | |
| | Sig. | 0.917 | 0.000 | – | | | | | |
| MI | ρ | 0.519 ** | 0.054 | 0.073 | – | | | | |
| | Sig. | 0.003 | 0.772 | 0.697 | – | | | | |
| ME | ρ | 0.063 | 0.272 | 0.299 | 0.157 | – | | | |
| | Sig. | 0.735 | 0.139 | 0.102 | 0.399 | – | | | |
| BI | ρ | 0.615 ** | 0.071 | 0.085 | 0.557 ** | −0.037 | – | | |
| | Sig. | 0.000 | 0.703 | 0.648 | 0.001 | 0.845 | – | | |
| A | ρ | 0.533 ** | −0.169 | −0.090 | 0.487 ** | −0.179 | 0.691 ** | – | |
| | Sig. | 0.002 | 0.364 | 0.632 | 0.005 | 0.335 | 0.000 | – | |
| SR | ρ | 0.089 | −0.247 | 0.048 | 0.268 | 0.422 * | 0.220 | 0.245 | – |
| | Sig. | 0.635 | 0.181 | 0.799 | 0.145 | 0.018 | 0.234 | 0.184 | – |

* Correlation is significant at the 0.05 level (two-tailed). ** Correlation is significant at the 0.01 level (two-tailed).

Quite a strong significant positive relationship was observed between the engineering and technology scores measured before the activity and the intrinsic situational motivation of the students ($\rho = 0.519$, $p = 0.003$). The engineering and technology scores were positively related to the intention to use microcontrollers ($\rho = 0.615$, $p < 0.001$) and the attitude toward similar activities and microcontroller programming in the future ($\rho = 0.533$, $p = 0.002$). The intrinsic motivation scores were significantly positively correlated with the scores of behavioral intention ($\rho = 0.557$, $p = 0.001$) and attitude towards future activities ($\rho = 0.487$, $p = 0.005$). Interestingly, extrinsic motivation scores correlated positively only with the project self-evaluation scores ($\rho = 0.433$, $p = 0.018$). Naturally, there was a direct relationship between behavioral intention to use microcontrollers and attitude towards future activities ($\rho = 0.691$, $p < 0.001$). Regarding technology anxiety, we saw a strong positive relationship between computer anxiety measured before the activity and microcontroller anxiety measured just after the activity ($\rho = 0.717$, $p < 0.001$).

Further exploration of the differences between groups of female and male students is presented in the next subsections.

### 3.3. Attitudes toward Engineering and Technology

In a whole group, the score for engineering and technology ranged from 16 to 45, with a mean score of 32.13 (Table 2). The differences between male and female students' attitudes towards engineering and technology are presented graphically in Figure 2, and these attitudes were higher in the subgroup of male students.

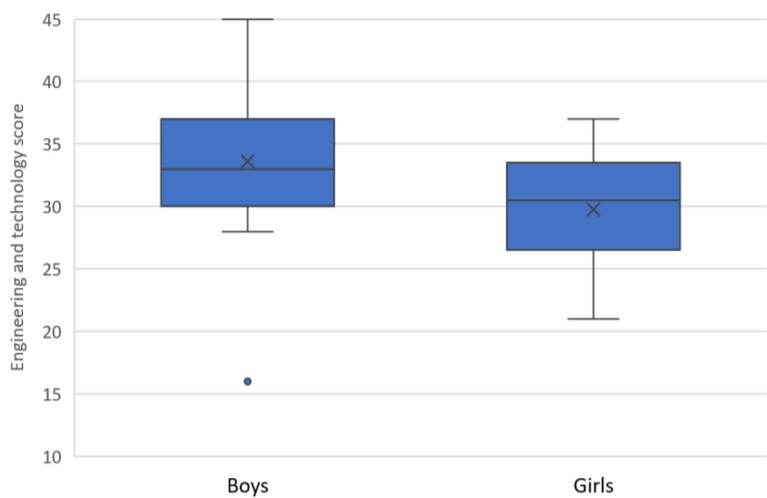

**Figure 2.** Differences in "engineering and technology" scores between male and female students (mean values are marked with ×).

The Mann–Whitney U test showed that the difference between the scores for engineering and technology in the subgroups of boys and girls was significant: U = 65.5, Z = −1.97, and $p = 0.048$, with boys' mean rank being 18.55 and girls' mean rank being 11.95.

### 3.4. Situational Motivation

We compared the scores for situational motivation using the intrinsic and extrinsic motivation sub-constructs in the whole group ($n = 31$) and in the groups of male ($n = 19$) and female ($n = 12$) students (Figure 3).

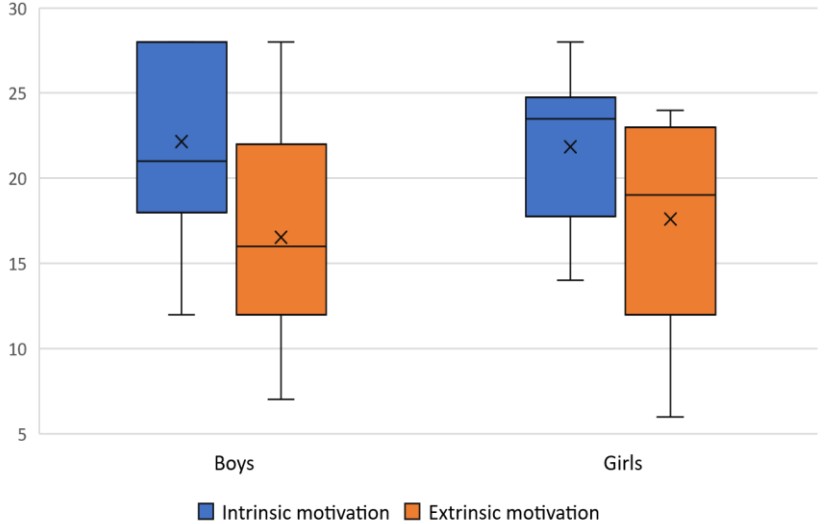

**Figure 3.** Differences in situational motivation scores between male and female students (mean values are marked with ×).

The Wilcoxon signed-rank test indicated that the intrinsic situational motivation for the group of all students was significantly higher than extrinsic motivation ($Z = -3.246$, $p = 0.001$).

The Mann–Whitney U tests for independent samples of boys and girls revealed no significant differences in situational motivation between male and female students (for intrinsic motivation, $U = 105.5$, $Z = -0.346$, $p = 0.729$; for extrinsic motivation, $U = 103.5$, $Z = -0.427$, $p = 0.669$; for general motivation, $U = 104.5$, $Z = -0.386$, $p = 0.699$).

### 3.5. Technology Anxiety

Technology anxiety (TA) was measured before the activity as computer anxiety and after the activity as "microcontroller anxiety", when students had become familiar with Arduino microcontrollers. For the whole group ($N = 31$), the computer anxiety measured before the activity was higher than the microcontroller anxiety measured after the activity if we assume the significance level of $\alpha = 0.1$ for the Wilcoxon signed-rank test ($Z = -1.894$, $p = 0.058$).

The results of the TA scores between the male and female groups are presented in Figure 4.

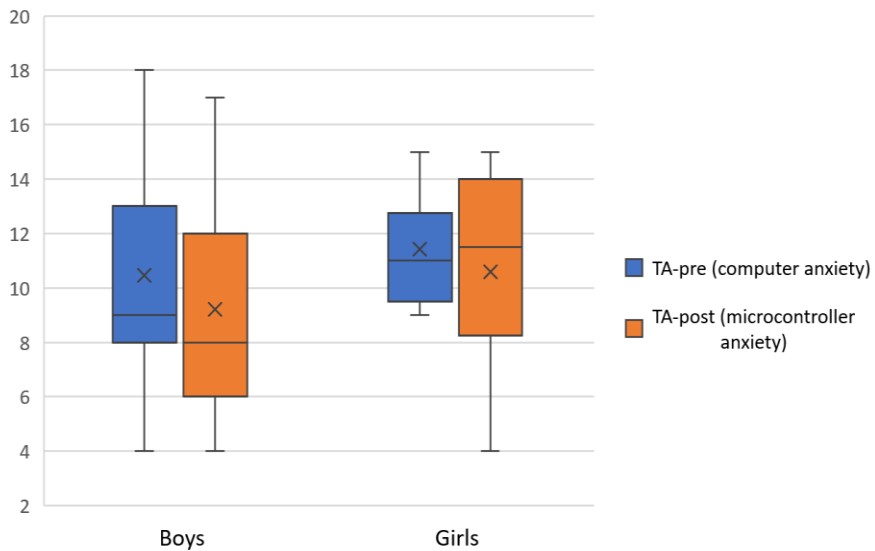

**Figure 4.** Differences in computer and microcontroller anxiety scores between male and female students (mean values are marked with ×).

We could see a slight decrease in the mean for microcontroller anxiety for the girls; however, this difference was not significant as confirmed by the Wilcoxon-signed-rank test ($Z = -0.923$, $p = 0.356$).

The decrease in technology anxiety for the boys was significant at the 0.1 level (Wilcoxon signed-rank test, $Z = -1.870$, $p = 0.061$).

### 3.6. Intentions toward Microcontrollers and Self-Evaluation

The scores for behavioral intention to use microcontrollers for the male and female students are depicted in a boxplot (Figure 5).

Based on the results of the Mann–Whitney U test ($U = 91.5$, $Z = -0.921$, $p = 0.357$), the differences in behavioral intentions towards microcontrollers were not significant for boys and girls.

Students' attitudes towards similar future activities with microcontrollers were measured. The mean scores for the whole group were high: 10.42 within the range of 3–15 (Table 2). However, there were no significant differences between boys and girls (Mann–Whitney U test, $U = 111$, $Z = -0.125$, $p = 0.900$).

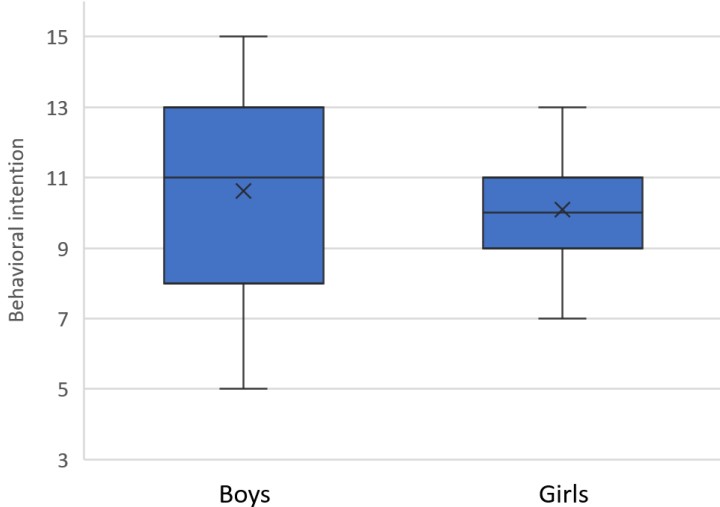

**Figure 5.** Differences in scores for behavioral intentions towards microcontrollers between male and female students (mean values are marked with ×).

On completion of the activity, students were asked to self-evaluate their projects, i.e., the results of their activity. The majority of the students evaluated them positively: "excellent" (41.9%) and "very good" (19.4%). As we can see in Table 3, there were no correlations between the behavioral intention and self-evaluation scores or between attitude towards similar future activities and self-evaluation scores. The Mann–Whitney U test confirmed that the differences between the groups of male and female students in their self-evaluations were not significant (U = 83.5, Z = −1.299, *p* = 0.194).

## 4. Discussion and Conclusions

Researchers have discussed efforts to improve and to increase students' interest and achievement in STEM education in a variety of ways. Our study focused on rural school students' individual characteristics and how they are associated with their engagement, gender differences, and inclusion in STEM by performing sustainable ecological engineering activities with Arduino microcontrollers. Our work also summarized the highlights from the literature on individual factors impacting engagement in STEM and gender stereotypes.

Based on the research results from Murphy [18], hands-on activities help to increase students' interest in STEM, and real-world learning increases the perceived utility value of STEM. The practical and operational nature of STEM education offers important opportunities to increase students' motivation and engagement [33].

In order to motivate and attract learners to STEM education, the educators in this study used hands-on physical computing projects related to a realistic context (the topic of ecology). The physical computing paradigm helps to bring computational concepts from the screen to the real world so that learners can interact with them [34].

Physical computing encompasses the creative arts and engineering design processes and combines hardware (e.g., sensors, LEDs, actuators) and software components [35]. While constructing Arduino-based prototypes, abstraction and logical thinking skills were needed to be used by the students in this study. Other computational thinking skills, such as algorithmic thinking, debugging, and modeling, were also needed during the implementation phase of the project. The implementation process of the project was based on principles that we thought would lead to success, as other studies often emphasized the importance of failure. Additionally, no prior theoretical background was given in order to provide students with the opportunity to learn the basics of physical computing and engineering in their integration with STEM subjects. Even if learners failed to solve a problem on an abstract level, they succeeded in doing so by using tangible objects. In such a way, students are motivated by their own creativity that is achieved by using microcontrollers [36], and a partial failure is an important part of the process, as it is

followed by understanding mistakes, tolerating disappointment, and encouraging students to discuss and debate. This might explain the findings of our study in that technology anxiety (in general and microcontroller anxiety in particular) had no negative correlation with students' situational motivation, intention to use microcontrollers, or attitudes toward similar microcontroller activities in the future. Students' initial attitudes toward engineering and technology fields had no negative relationship with technology anxiety either.

Although the literature suggests that literacy levels are lower in rural schools and that there is a lack of qualified teachers and of adequate provision of technology [18], the results of our study showed that students were successful in implementing projects and that intrinsic motivation was encouraged. Even occasional activities, such as short-term interventions in education, would help to increase students' interest in engineering and technology and girls' involvement. The inclusion of such activities would help to attract students to engineering fields.

Students' academic emotions and motivational beliefs are linked to their engagement and participation in STEM subjects, as well as to their long-term STEM career choices, regardless of their abilities and prior achievements [37].

### 4.1. Inter-Relationships Students' Technological Attitudes and Behavioral and Motivational Factors

Our study shows that students with a more positive attitude toward engineering and technology are more intrinsically motivated by microcontroller activities. These students are also more likely to use microcontrollers in the future and express stronger wishes for similar activities.

Technology anxiety when using microcontrollers was naturally found to have a positive association with the initial general computer anxiety observed in the students before the activity. Many studies have confirmed the significant direct or indirect negative effect of technology anxiety on intention to use technology (e.g., [38–41]). In the context of our study, which was held in a school in a rural area among students whose initial anxiety level might naturally be higher than that of students in urban schools due to the worse technology availability in schools and at home, our study showed that technology anxiety levels did not directly predict students' intentions to use microcontrollers and to be involved in similar activities in the future. This may also be related to the relevance of the topics of the integration of microcontroller activities in the context of ecology and agriculture, the design and methodology of activities, and the emphasis on the importance of partial failure in the learning process, as discussed. Within the whole group, the students were significantly more intrinsically than extrinsically motivated by the microcontroller activities. This suggests that microcontroller activities seem relevant and interesting to students despite their technology anxiety levels, and the students are interested by the content and context of the activity itself. A study by Jungert et al. [42] confirmed that intrinsic motivation predicted students' intentions to persist in STEM and select future enrollment in a college STEM program. The microcontroller activities provoked intrinsic situational motivation, which suggests implications for developing possibilities for future STEM studies and career choices by students.

Our study found that the intrinsic motivation of the students was not related to their self-evaluation of the results of the completed activity. This is in line with the results of [42], where no significant paths were detected in the model between students' intrinsic motivation and achievement. It is likely that more studies can be found in the literature that prove this finding, such as [43], the authors of which believe that successful and effective learning is encouraged by physical computing; they also observed that physical computing activities resulted in higher motivational values in the computer science classroom. However, our study showed that students whose situational motivation appeared to be extrinsic self-scored their activity results higher, which suggests that extrinsically motivated students might be looking for external stimuli, e.g., possible praise for a high-scoring project.

Our findings show that the use of microcontrollers in programming class had a positive effect on students' attitudes toward microcontrollers, as well as an increase in students' self-efficacy, which supports the results reported in [44]

### 4.2. Gender Differences in Individual Factors Studied

Gender differences in the use of technology have been reported in many studies, including those on physical computing, e.g., [45]. Previous studies also provided results showing that males' abilities in explaining connections between computing concepts were rated significantly higher than those of females [9].

Similarly to the finding of [27] that women perceive themselves as having less advanced technological self-efficacy skills compared to men, our study confirms that female students' attitudes towards engineering and technology are significantly less positive than those of male students.

In line with a previous study [46] where exposure to female STEM experts was provided in order to change female participants' outlook on work–life balance, we think that one of the possible factors for why the motivation of girls for taking an interest in STEM after the activities may have been that the activities were led by women from STEM and the engineering sciences. The lecturers told their life stories and described their paths and the obstacles they faced in their profession.

Additionally, we believe that the chosen gender-neutral, inclusive ecology topic may also have stimulated students' interest in STEM, especially for female students. As suggested by [5], for women, it is important to emphasize a sense of community utility in STEM fields.

Threats of gender-based stereotypes about how females should think and feel about STEM-related activities and outcomes might be related to higher levels of learning anxiety for female students [47], and, in turn, anxiety related to learning STEM subjects has a stronger negative effect on self-efficacy for females than for males [42].

Real-life projects with microcontrollers help to increase students' technological acceptance, as reported in [48], a study on maker education activities that improve students' self-efficacy in relation to their belief of being good at technology and science, especially among girls. In our study, a significant (at the 0.1 level) decrease in technology anxiety after completing the activity was observed for the male students. However, as technology anxiety levels (before and after the activity) were not directly associated with attitudes toward similar activities and students' intentions to use microcontrollers, this might suggest that real-world microcontroller activities involve girls and help them overcome existing technology anxiety. In addition, other researchers suggested that physical computing activities enhance the computational thinking self-efficacy of female students [49].

### 4.3. Limitations and Directions for Further Research

The limitation of our work is that the activities were performed with a small sample and short interventions. However, small-sample findings can be useful for future studies that include larger sample sizes and longer-term interventions for deeper engagement in STEM that have positive effects on participants [1]. The further directions provided by our study are to draw larger samples and conduct longer-term interventions in order to compare changes in students' motivation and attitudes toward engineering, technology, and other STEM subjects. Future research should also identify what impact occasional activities in rural schools could have on making the area of STEM attractive for future careers.

**Author Contributions:** Conceptualization, G.S. and A.J.; methodology, T.J.; formal analysis, T.J.; investigation, G.S. and A.J.; resources, A.J.; data curation, T.J.; writing—original draft preparation, G.S., A.J. and T.J.; writing—review and editing, G.S., A.J. and T.J.; visualization, T.J.; project administration, G.S.; funding acquisition, G.S. All authors have read and agreed to the published version of the manuscript.

**Funding:** The APC was funded by Vilnius University.

**Institutional Review Board Statement:** Not applicable.

**Informed Consent Statement:** Informed consent was obtained from the responsible authorities involved in the study.

**Data Availability Statement:** Original data are available on request. They are not archived in any repository.

**Acknowledgments:** Data collection was conducted under the STEAM-CT project funded by the Erasmus+ Program (agreement No. 2019-1-BE02-KA201-060222).

**Conflicts of Interest:** The authors declare no conflict of interest.

### Appendix A

To manage a smart greenhouse, we used an Arduino kit with particular actuators and sensors, which were used to detect the useful data in order to grow plants and identify the necessary modifications to the environment.

**Table A1.** The smart greenhouse projects.

| Projects | Sensors | Actuators | Scheme |
|----------|---------|-----------|--------|
| The temperature and humidity of the air | DHT11-HUMIDITY AND TEMPERATURE to detect the temperature and humidity of the air inside the greenhouse | LEDs that will turn on or off depending on the data detection levels. Possibility for development by inserting an LCD DISPLAY with an I2C driver |  |
| The concentration of a gas | GAS SENSOR to detect the concentration of a gas (carbon dioxide) | BUZZER to generate a sound and an LED that will light up when above a gas concentration threshold |  |
| Brightness level inside the greenhouse | PHOTORESISTANCE for solar brightness control | LED that will light up when below a brightness threshold |  |

**Table A1.** *Cont.*

| Projects | Sensors | Actuators | Scheme |
|---|---|---|---|
| Control of the light with a mobile device | ON/OFF LED to illuminate the greenhouse when needed | BLUETOOTH | 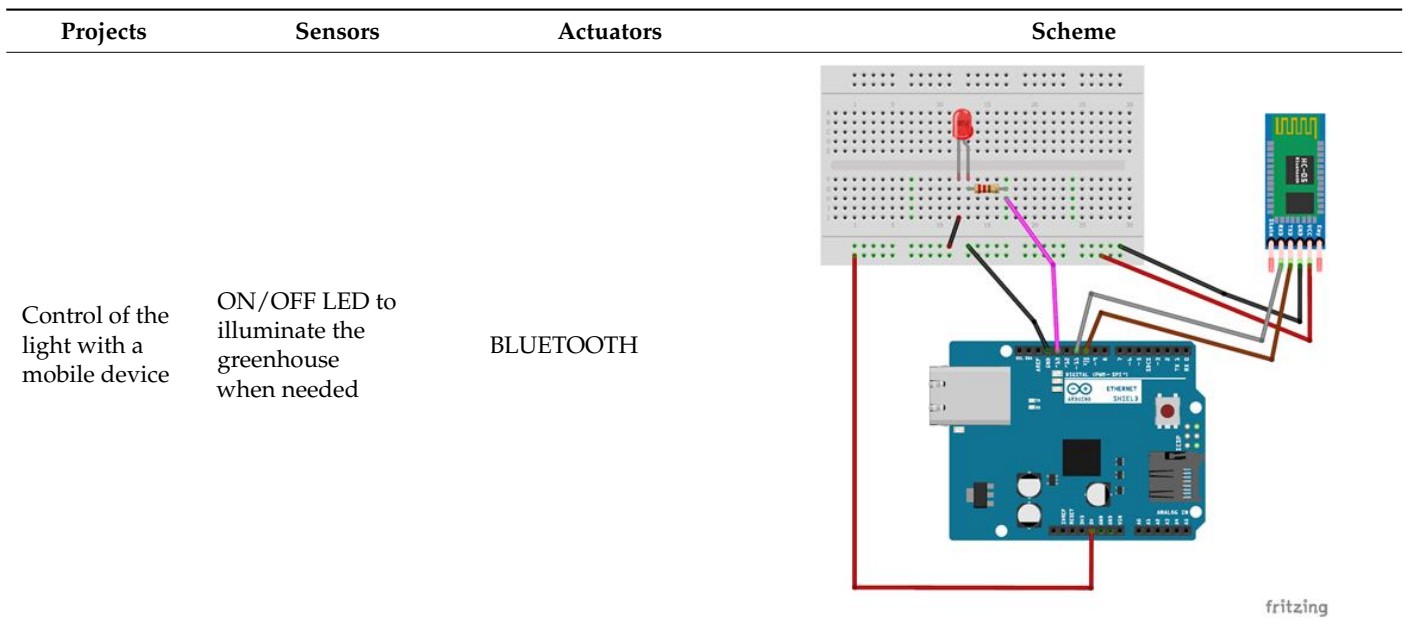 |

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
