# Peer review of "Solving Ecological Problems through Physical Computing to Ensure Gender Balance in STEM Education"

_sustainability, doi:10.3390/su14094924_

Round 1

Reviewer 1 Report

The organization of the article would be more appropriate if the section "1. Introduction and Background" was divided into two sections: "1. Introduction" and "2. Background and Motivation".

In section 1 was presented the following justificative to use Arduino: "Activities with Arduino micro- controllers were chosen as they were designed to engage children in several different aspects of engineering including: building, programming, designing, and iterative testing and redesign [11,12] and have a positive effect on students’ academics and understanding of engineering concepts...". This justificative should be better detailed. It is necessary to describe what are the advantages and limitations of using the Arduino in relation to other microcontrollers.

There is an error in table 1. It was listed that among the 31 participants 12 are Female  and 19 Male. The correct percentage (%) is of 38.7% Female and 61.3 for Male.  In the table the percentages are inverted.

As most of the research participants had not had previous contact with microcontrollers, why was it not considered using Tinkercad to perform simulations before using Arduino? Couldn't that reduce students' anxiety?
One other point. It's just that Tinkercad is a free online platform. Students could use it from their homes to continue performing virtual simulations and experiments even without the Arduino board and sensors and actuators. Couldn't this be a way to motivate them and stimulate their interest in STEM subjects?

In the list of references, the  reference 1 was entered as:  "Author 1, A.; Author 2, B. Book Title, 3rd ed.; Publisher: Publisher Location, Country, 2008; pp. 154–196". Please enter the correct data for this reference or delete it and update the reference list.

Author Response

Response to Reviewer’s comments.

REVIEW 1

  1. The organization of the article would be more appropriate if the section "1. Introduction and Background" was divided into two sections: "1. Introduction" and "2. Background and Motivation".

Response: We rearranged structure of the Section 1. We have changed the title to “1. Introduction” and added the subsections: “1.1. Background and motivation” and “1.2. Aim and research questions”.

  1. In section 1 was presented the following justificative to use Arduino: "Activities with Arduino micro- controllers were chosen as they were designed to engage children in several different aspects of engineering including: building, programming, designing, and iterative testing and redesign [11,12] and have a positive effect on students’ academics and understanding of engineering concepts...". This justificative should be better detailed. It is necessary to describe what are the advantages and limitations of using the Arduino in relation to other microcontrollers.

Response: The text was improved by several sentences:

In this research, we introduced sustainable ecology (smart greenhouse) problems for students from rural lower secondary school (grade 7–8). Activities with Arduino microcontrollers were chosen as they were designed to engage children in several different aspects of engineering including: building, programming, designing, and iterative testing and redesign [11,12] and have a positive effect on students’ academics and understanding of engineering concepts [13], also fostering students’ creativity and a sense of discovery [14] that are important problem-solving skills [15]. Arduino was chosen based on its functionality, suitable for the activity design, availability and previous positive experience in Arduino-based teaching activities. The Arduino prototyping board is considered to be an attractive physical device that does not require a lot of time to learn how to use it, how to configure it, and is therefore becoming increasingly popular among researchers and in educators. In fact, the selection of a particular model of a microcontroller is not essential, as the goal is to understand the main principles of microcontroller programming to solve the problem posed. When the main principles are learned, the learning can be transferred and applied with another technological tool.

It is important to point out that our activities at rural school were led by a female teaching team, which have previous experience in attracting girls into STEM field by activities with Arduino. Sullivan and Bers’s [16] study has shown preliminary evidence that female robotics instructors may have a more positive impact on girls’ performance on robotics tasks. Female mentor role models are seen as one of the promising ways to reduce gender gap in STEM [17].

  1. There is an error in table 1. It was listed that among the 31 participants 12 are Female and 19 Male. The correct percentage (%) is of 38.7% Female and 61.3 for Male. In the table the percentages are inverted.

Response: Done.

  1. As most of the research participants had not had previous contact with microcontrollers, why was it not considered using Tinkercad to perform simulation before using Arduino? Couldn't that reduce students' anxiety? One other point. It's just that Tinkercad is a free online platform. Students could use it from their homes to continue performing virtual simulations and experiment seven without the Arduino board and sensors and actuators. Couldn't this be a way to motivate them and stimulate their interest in STEM subjects?

Response: The text was improved by adding several sentences:

In 2. Materials and Methods:

The initial idea was to use Tinkercad for greenhouse modelling before using Arduino, but there was no such extensive capability because Tinkercad lacked several of the necessary libraries and sensors. On the other hand, the students had just come back from distance learning, so from a practical point of view they were more lacking in real simulation and the possibility to see how the sensors work in a real environment.

  1. In the list of references, the reference 1 was entered as: "Author 1, A.; Author 2, B. Book Title, 3rd ed.; Publisher: Publisher Location, Country, 2008; pp. 154–196". Please enter the correct data for this reference or delete it and update the reference list.

Response: deleted, all references were re-numbered and cited without this first reference.

Reviewer 2 Report

Dear author(s), I read your article with interest related to gender stereotypes and technological self-efficacy. The paper is well written with adequate referred articles. The discussion and conclusions fundament the research questions and provide information for reflection. Including the questionnaires designed -and adapted- for the statistical analysis of the responses in the annex would have been illustrative.

Author Response

Response to Reviewer’s comments

REVIEW 2

Dear author(s), I read your article with interest related to gender stereotypes and technological self-efficacy. The paper is well written with adequate referred articles. The discussion and conclusions fundament the research questions and provide information for reflection. Including the questionnaires designed -and adapted- for the statistical analysis of the responses in the annex would have been illustrative.

Response: thank you very much for positive review! The links to the initial instruments used in the study are provided in section 2.2. The survey and instruments.

Reviewer 3 Report

The paper deals with an interesting and up-to-date issue (gender differences in motivation for STEM education). It is well written and provides new results.

Some things have to be corrected (see below).

Critical remarks:

Methodology

In my opinion, RQ2 is not precise enough:
RQ2. What are the differences between groups of female and male students in terms of individual characteristics studied?
(what are "individual characteristics studied"? - please specify)

You declare that "The study [44]" "proves our findings". But that study is from 2018, the authors did not know your findings, so how could they prove them? It's the opposite: you could prove their findings, but also this is not true, as I cannot see any mathemtical proofs in your paper. I'd rather say that your results are supporting the [44] ones.

Important information about the experiment is missing.
When exactly the experiment took place and how long it lasted?
Were the students informed ahead of the experiment that they were taking part in it?

Detailed remarks

Some things are unclear. For instance, "Codes were slightly modified..." - by whom? Teachers? Students?

And how old are "7–8 grade students" in Lithuania?

Language

Proofreading is suggested as small language errors were noticed, e.g.:
In many professional STEM (science, technology, engineering, and math) fields the
=>
In many professional STEM (science, technology, engineering, and math) fields, the

Style:
Numbers should not begin sentences as in: "10% significance..."
Avoid colloquial phrases ("a rather high mean").

Author Response

 Response to Reviewer’s comments

REVIEW 3

The paper deals with an interesting and up-to-date issue (gender differences in motivation for STEM education). It is well written and provides new results. Some things have to be corrected (see below).

Critical remarks: Methodology

  1. In my opinion, RQ2 is not precise enough: RQ2. What are the differences between groups of female and male students in terms of individual characteristics studied? (what are "individual characteristics studied"? - please specify)

Response: individual characteristics has been added in RQ2.

  1. You declare that "The study [44]" "proves our findings". But that study is from 2018, the authors did not know your findings, so how could they prove them? It's the opposite: you could prove their findings, but also this is not true, as I cannot see any mathematical proofs in your paper. I'd rather say that your results are supporting the [44] ones.

Response: the sentence has been rewritten:

Our findings show that the use of microcontrollers in programming class had a positive effect on students' attitudes toward microcontrollers as well as an increase in students’ self-efficacy, what supports the results reported in [44].

  1. Important information about the experiment is missing. When exactly the experiment took place and how long it lasted? Were the students informed ahead of the experiment that they were taking part in it?

Response: information about experiment is provided in section 2. Materials and methods and missing information has been added:

Students were informed in advance about the survey they were taking part in, and that their participation is voluntary.

  1. Some things are unclear. For instance, "Codes were slightly modified..." – by whom? Teachers? Students?

Response: Codes were slightly modified by instructors of activities to adapt to the classroom environment as the temperature and brightness level in different rooms varied.

  1. And how old are "7–8 grade students" in Lithuania?

Response: we have added age information in section 2. Materials and methods. It is also provided in section “3. Results” and in Table 1.

  1. Proofreading is suggested as small language errors were noticed, e.g.: In many professional STEM (science, technology, engineering, and math) fields the=>In many professional STEM (science, technology, engineering, and math) fields, the

Response: Done

  1. Style: Numbers should not begin sentences as in: "10% significance..."Avoid colloquial phrases ("a rather high mean").

Response: Thank you, we have changed this in several sentences where such cases appeared.
